# Motivation and Basic Psychological Needs Satisfaction in Active Travel to Different Destinations: A Cluster Analysis with Adolescents Living in Germany

**DOI:** 10.3390/bs13030272

**Published:** 2023-03-20

**Authors:** Denise Renninger, Joachim Bachner, Xavier García-Massó, Javier Molina-García, Anne Kerstin Reimers, Isabel Marzi, Franziska Beck, Yolanda Demetriou

**Affiliations:** 1Department of Sport and Health Sciences, Technical University of Munich, 80992 Munich, Germany; 2AFIPS Research Group, Department of Teaching of Physical Education, Arts and Music, University of Valencia, 46022 Valencia, Spain; 3Epidemiology and Environmental Health Joint Research Unit, FISABIO-Universitat Jaume I-Universitat de València, 46020 Valencia, Spain; 4Department of Sport Science and Sport, Friedrich-Alexander-Universität Erlangen-Nürnberg, 91058 Erlangen, Germany

**Keywords:** active transport, commuting, youth, health promotion, behavioral regulation, ARRIVE

## Abstract

Active travel in adolescence contributes to improved health outcomes. Self-Determination Theory suggests that motivation and basic psychological needs influence travel behavior. Person-centered approaches can examine interrelationships of these constructs underlying travel behavior. The aim of this study was to investigate (i) which clusters can be identified in adolescents, (ii) whether clusters explain overall active travel behavior, (iii) whether clusters were associated with travel mode to various destinations or distance, and (iv) whether differences across clusters appear regarding sex/gender, age, and weight status. The sample included 517 (263 male, 254 female) adolescents from Germany, aged 11–15. Self-organizing maps analysis identified six clusters from nine input variables: intrinsic motivation, integrated regulation, identified regulation, introjected regulation, external regulation, amotivation, autonomy satisfaction, competence satisfaction, and relatedness satisfaction. The most beneficial cluster regarding active travel demonstrated highest basic psychological needs satisfaction and autonomous motivation with low controlled motivation and amotivation. The most vulnerable cluster was characterized by generally low levels of motivation except for external regulation and amotivation. Clusters were not associated with distance to school, friends/relatives, shopping facilities, or leisure facilities. The findings support the importance of high quality and high quantity of motivation for active travel in adolescents.

## 1. Introduction

Engaging in regular physical activity (PA) in youth promotes overall health and contributes to the prevention of non-communicable diseases [1,2,3]. Thus, the global action plan calls for more active people and reduced physical inactivity [4]. However, PA levels among adolescents are alarmingly low worldwide [5,6]. This also applies to Germany, with only four percent (device-base measured PA) or nine percent (self-reported PA) of children and adolescents adhering to the WHO recommendations of 60 min moderate-to-vigorous PA per day [7]. One opportunity to integrate PA on a daily basis is to use an active travel (AT) mode such as walking or cycling to reach destinations [4]. Research emphasizes the importance of AT to increase PA in adolescents [8,9]. Furthermore, adolescents who travel actively can benefit from improved psychological well-being [10,11], physical fitness [9,12], and emotional health [13]. Besides this, AT is a sustainable and environmental-friendly way of traveling [14]. However, AT is declining in many countries all over the world [4]. In Germany, only a minority of adolescents regularly walk or cycle to destinations such as school [15,16].

This lack of AT in adolescence emphasizes the need for interventions. In order to develop successful interventions, a thorough understanding of modifiable factors and mechanisms of AT is needed. Self-determination theory (SDT) is a widely acknowledged motivational framework for psychological processes and factors that underlie a certain behavior [17,18]. SDT suggests that not only the quantity of motivation but also the quality determines whether a behavior is performed or not [18]. SDT conceptualizes motivation as a construct with two general forms of motivations: intrinsic and extrinsic motivation [17,18,19,20]. Intrinsic motivation represents the most autonomous (or self-determined) form of motivation stemming from the inherent pleasure and enjoyment of performing the behavior [18,19]. Extrinsic motivation refers to behaviors that are performed for external reasons rather than for the inherent satisfaction caused by performing the behavior [17,20]. Based on these various (extrinsic) reasons, extrinsic motivation is further distinguished. The most autonomous form of extrinsic motivation is integrated regulation, which means that the person’s identity, personal values, and goals align with performing the behavior. A less self-determined (but still autonomous) form is identified regulation, which originates from the importance a person attributes to the behavior. Introjected regulation is defined as a more controlled form of extrinsic motivation, where the behavior is driven by internal pressure to avoid guilt or to improve one’s self-esteem. The most controlled form is external regulation, which results from external pressure from other people in order to obtain rewards or to avoid punishments. SDT arranges these different forms of motivation along the continuum of self-determination [17,20]. Decreasing in their degree of self-determination, the three autonomous forms of motivation (intrinsic motivation, integrated regulation, and identified regulation) are followed by the controlled forms (introjected regulation and external regulation). At the non-self-determined end of the continuum, amotivation represents a complete lack of motivation or intention to engage in the behavior [17,20].

SDT proposes that in order to foster self-determination, the satisfaction of the three basic psychological needs (BPNs), autonomy, competence, and relatedness, is fundamental [17,18]. A more self-determined form of motivation increases the likelihood of performing or maintaining a certain behavior. In PA settings, previous research concerning adolescents supports the positive association of autonomous motivation and PA or exercise [21,22] as well as the positive effect of BPNs satisfaction on PA [23,24]. Despite these positive effects, SDT has only recently gained recognition in the context of AT. Specifically, research from Spain, Portugal, and Sweden supports the applicability of SDT in the context of AT to/from school [25,26,27,28,29]. Intrinsic motivation, integrated regulation, and identified regulation positively predicted active school travel in Spanish secondary students [25]. The research from Sweden identified similar positive associations of theses autonomous forms of motivation with active school travel and an additional negative association of amotivation with active school travel [29]. However, SDT-based research in relation to different destinations other than school is lacking. Furthermore, theses aforementioned studies provide good examples of the variable-based approaches traditionally used in SDT-based research. Such approaches examine relationships between motivation or BPNs and a behavior (e.g., [21,30,31,32,33]). With regard to motivation, this typically involves pitting the different forms of motivation against each other to identify the most beneficial or detrimental one for a specific behavior. By doing so, variable-based approaches are unable to fully incorporate SDT’s theoretical conceptualization of motivation, which includes recognizing that behavior can be motivated by multiple reasons simultaneously [20]. Thus, they neglect the notion of the co-existence of different forms of motivation within groups of individual to varying degrees [34]. This supports the call for more person-centered research such as cluster analyses [35]. These kind of approaches can depict the heterogeneity of motivation by capturing the interplay of the single forms within a person [35,36]. Thus, profiles of naturally occurring combinations of different motivational types within groups of people can be identified, which illustrate specific motivational patterns underlying a behavior [34,35,36]. Given the theoretically proposed simplex structure of motivation [18,37], a profile with a single dominant form of motivation combined with some endorsement of proximate forms and decreasing endorsement of more distant ones (i.e., displaying an unimodal curve) would be more common than profiles that exhibit matching levels of opposing forms of motivation [36]. Based on SDT, research has been conducted aiming to identify profiles representing groups of people who share certain motivational characteristics regarding, for example, motivation for physical education [38,39,40], physical activity [41], or exercise [42,43]. Throughout these studies, research identified various clusters that comprised distinct combinations of the different motivational types. This included both clusters that somewhat followed theoretical assumptions by demonstrating either higher levels of controlled and lower levels of autonomous motivation or vice versa and clusters that were characterized by comparable levels in autonomous and controlled motivation.

Person-centered research has also been conducted to identify profiles concerning BPNs satisfaction [41,44,45]. Considering the theoretical perspective that autonomy, competence, and relatedness constitute key nutriments of high quality motivation [18,20], profiles with high BPNs satisfaction would be observed in combination with higher autonomous motivation. Agreeing with SDT, research from physical education found higher autonomous motivation in clusters of students with high BPNs satisfaction compared to clusters of lower BPNs satisfaction [41,45]. However, in clusters of university students, the cluster characterized by mainly extrinsic motives for PA exhibited higher levels of autonomy satisfaction than the cluster of intrinsic motives [44].

These findings, once again, exemplify the ability of person-centered analyses to capture the diverse composition of subgroup characteristics that could not be fully captured using variable-based analyses. More broadly, this enables the identification of particularly vulnerable and resilient groups, and thus can contribute to the production of more tailored intervention programs [35]. With regard to the debate on the heterogeneity of motivation, the person-centered approach may be specifically helpful to gain a better understanding of the motivational mechanisms underlying AT in adolescents. To the best of our knowledge, no previous study has examined SDT-based clusters of AT behavior in adolescents. Thus, this research aims to answer the following research questions: Which clusters comprising motivation and BPNs satisfaction regarding AT behavior can be identified in adolescents? To gain a better understanding of the co-existence of the psychological determinants of AT, clusters will be generated using self-organizing maps (SOM) analysis. The resulting profiles will be evaluated based on the theoretical assumptions of SDT [17,18].Does overall AT behavior differ between the generated clusters? Current variable-based research indicates that both motivation and BPNs satisfaction affect AT behavior [25,26]. Thus, we assume that, if the SOM analysis results in theory-consistent profiles, differences in overall AT behavior between the clusters will be observable. Overall AT behavior should be higher in clusters with higher levels of autonomous motivation and higher BPNs satisfaction compared to clusters with higher levels of controlled motivation or amotivation and lower levels of BPNs satisfaction.Is travel mode or travel distance associated with cluster membership when considering different destinations? Most research on travel behavior has been conducted on travel behavior to and from school [8]. However, adolescents frequently travel to other destinations in everyday life, such as to shopping or leisure facilities [46]. Therefore, we aim to examine whether an association occurs between the number of adolescents who choose an active travel mode and their cluster membership by considering different destinations. Additionally, distance to a destination has been found to strongly influence travel mode choice [47,48,49]. Accordingly, we will investigate whether the distance traveled to a certain destination differs between clusters and whether this might contribute to explain travel mode choices.Are there differences across the clusters in terms of sex/gender, age, and weight status? Sex/gender and age predicted cluster membership in a study on motivational profiles in physical education [50]. Additionally, previous research illustrates lower levels of AT in older youth [15] and female adolescents [16]. Further, research suggests a negative association between AT behavior and body-mass-index (BMI) [51,52]. Thus, differences in the distribution of male and female adolescents, age, and weight status might also occur across the clusters.

## 2. Materials and Methods

### 2.1. The ARRIVE Study

The present research is conducted within the ARRIVE (Active tRavel behavioR in the famIly enVironmEnt) study. This cross-sectional study aimed to gain a deeper understanding of the social and individual factors that influence adolescents’ AT behavior [53]. The study addressed several gaps of current research in AT among youth. Specifically relevant for the present investigation is that ARRIVE focused on travel behavior, including more destinations instead of being restricted to the school domain like most prevailing research [8,53]. Additionally, it addresses the population of adolescents, which is rather under-researched compared to younger children [8,54,55]. A detailed description of the ARRIVE study and all theoretical underpinnings can be found in the study protocol [53]. By exploring SDT-based clusters in the adolescent population, the present study contributes to the overall aim of the ARRIVE study to gain a deeper understanding of the diverse factors involved in the decision-making process on travel mode choice. 

### 2.2. Study Participants

The target population included adolescents aged 11–15 years and one of their parents. In total, 517 parent-adolescent dyads comprise the study’s population. In total, 263 boys and 254 girls with a mean age of 13.11 years (male: M_age_ = 13.21 ± 1.33; female: M_age_ = 12.92 ± 1.35) constituted the study sample. Adolescents were from cities with more than 100,000 inhabitants (29.2%), medium-sized towns consisting of 20,000–99,999 inhabitants (17.4%), small towns with 5000–19,999 inhabitants (22.2%), and living areas with less than 5000 inhabitants (30.8%) in Germany. Adolescents’ average BMI was 19.23 kg/m^2^ (SD = 3.32, N = 405, min = 12.48, max = 34.36). For some adolescents, height or weight was not available, thus BMI was calculated for 405 participants. 

### 2.3. Data Collection

The survey took place in June 2021. The sample was drawn from a nation-wide online panel (forsa.omninet), which is representative of the German population with regard to age, gender, education, and place of residence. The study collected data from adolescents and one of their parents. Parents were recruited purposively, aiming at an approximately equal number of fathers and mothers and approximately equal number of male and female adolescents aged 11–15 years. Parents were initially contacted by telephone, asking if they were interested in participating in the study. Afterwards, they received detailed information on the study aim and a link to the online questionnaire via e-mail. Prior to study participation, parents and adolescents were asked for their consent to participate. Only if consent was provided, participants were able to start the survey. The questionnaire was structured in two sections: parents completed the first part of the questionnaire. After that, parents were asked to let the adolescent fill out the second part on his or her own. 

### 2.4. Measures

#### 2.4.1. Sociodemographic Data

Socio-demographic data was obtained from the parental section of the questionnaire. Parents indicated adolescents’ sex/gender, age, height, and weight.

#### 2.4.2. Motivation towards Active Travel

To assess adolescents’ motivation towards AT, the items from the Spanish Behavioural Regulation in Active Commuting to and from School (BR-ACS) Questionnaire [25] were translated into German. Additionally, items were adapted to focus on motivation towards AT in general (no exclusive focus on AT to/from school). Thus, the expression “to and from school” was replaced with “cover a distance”. The translation process commenced with the translation of the original BR-ACS by two independent researchers and the discussion of the resulting two translations. After reaching consensus, the items were reviewed by three experts in the field of youth activity behavior and SDT and subsequently retranslated to ensure the consistency of the German items with the original items. The resulting questionnaire was given to four adolescents (two boys, two girls) to evaluate the acceptability and understanding of the questionnaire. Their feedback resulted in the final German version of the BR-ACS which was used in the study.

The questionnaire comprised six subscales that aimed to assess adolescents’ intrinsic motivation (four items; e.g., “I cover distances by foot or bike because it’s fun”), integrated (four items; e.g., “I cover distances by foot or bike because it fits to who I am”), identified (three items; e.g., “I cover distances by foot or bike because I value the advantages”), introjected (four items; e.g., “I cover distances by foot or bike because I feel guilty when I don’t do so”) and external regulation (four items; e.g., “I cover distances by foot or bike because other people say I should do so”), and amotivation (four items; e.g., “I don’t see the sense of covering distances by foot or bike”) regarding AT. Participants responded on a 5-point Likert scale, from 0 (do not agree at all) to 4 (completely agree). For each subscale, a mean value was calculated for each participant. Previous research demonstrated an appropriate fit of the 6-factor model representing the six motivational types in Spanish, Portuguese, and Swedish adolescents, and supported the simplex structure proposed by SDT [25,27,29]. In the present sample, internal consistency coefficients ranged from 0.69 to 0.91.

#### 2.4.3. Basic Psychological Needs Satisfaction towards Active Travel

Adolescents’ BPNs satisfaction was assessed using an adapted version of the Spanish Basic Psychological Need Satisfaction in Active Commuting to and from School (BPNS-ACS) Questionnaire [26]. The items were translated into German and adapted in the same way as it was done with the motivational questionnaire. The questionnaire comprised three subscales that aimed to assess adolescents’ autonomy (four items, e.g., “I can choose how I cover distances”), competence (four items, e.g., “I feel that I have the necessary skills to cover distances on foot or by bicycle”), and relatedness (four items, e.g., “I feel very comfortable with the people who accompany me”) satisfaction regarding AT. The twelve items were answered using a 5-point Likert scale, from 0 (do not agree at all) to 4 (completely agree). For each of the three subscales, a mean value was calculated for each participant. Previous research showed an acceptable fit of the 3-factor model (autonomy, competence, and relatedness) and found that the questionnaire was valid and reliable in a sample of adolescents [26,27]. For the present study, internal consistency coefficients for autonomy, competence, and relatedness satisfaction were 0.75, 0.85, and 0.76, respectively.

#### 2.4.4. Travel Behavior

Adolescents’ travel mode (e.g., by bike, by bus, by car, etc.) was assessed for relevant destinations in adolescence [46]: to school, from school, to friends/relatives, to shopping opportunities, and to leisure facilities. The assessment was based on the Mode and Frequency of Commuting To and From School Questionnaire [56] and the Mobility in Germany survey (Mobilität in Deutschland, MiD, [57]). The given answers regarding the usual travel mode on the way to the respective destinations were subsequently categorized into either active (e.g., walking, cycling) or passive (e.g., being driven by car, using public transport). Thus, we obtained a dichotomous variable (active or passive) for each destination. Additionally, travel distance to each destination was assessed. Parents reported the distance to school. Distance to shopping facilities, friends/relatives, and leisure facilities was obtained from adolescents’ self-reports. Adolescents were asked to indicate the distance to the respective destinations according to the following categories: less than 500 m, between 500 m and 1 km, between 1 km and 2 km, between 2 km and 3 km, between 3 km and 5 km, and more than 5 km.

To depict adolescents’ overall travel behavior, we first calculated the total number of destinations a participant usually reached actively and the total number of destinations reached passively. The adolescents were also asked to report whether they usually visit the different destinations at all. In a next step, we calculated the ‘proportion of ways traveled actively’ by dividing the total number of destinations a participant usually reached actively by the sum of all destinations a participant reported to travel to. 

### 2.5. Data Analysis

SOM analysis was used to classify participants into clusters and provide profiles according to participants’ similarities regarding the input variables [58]. The nine variables intrinsic motivation, integrated regulation, identified regulation, introjected regulation, external regulation, amotivation, autonomy satisfaction, competence satisfaction, and relatedness satisfaction were used as input variables. The SOM analysis was conducted with the Matlab R2018a program (Mathworks Inc., Natick, MA, USA) and the SOM toolbox (version 2.0 beta) for Matlab [59]. The procedure to obtain the SOM contained three steps [58]: network building, initialization, and training. During network building, a neuron network was constructed, by selecting the lattice size of 14 × 8 neurons based on the sample size of this study. Thereby, each neuron is represented by an empty vector with length equal to the number of input variables. Second, during the initialization, a starting weight (-value) is assigned to each neuron. This is performed for each input variable and in two different ways: randomized and linear initialization. In the third step, two training algorithms (i.e., sequential and batch) were applied to modify the initially assigned weights of the neurons [60]. The modification of the neuronal weights in each iteration of the training process is influenced by several factors. After an input vector (representing a study participant’s characteristics in the input variables) is introduced to the neuron network, the neurons compete in order to win the input vector. The neuron with the smallest Euclidean distance to the input vector is the winning neuron. Thus, the winning neuron exhibits the weight vector with the closest values to the input vector’s values. Subsequently, all neurons in the network adapt their weight values to the values of the input vector [61]. A neuron’s ability to adapt, in turn, depends on the learning ratio and the neighborhood function. The learning ratio commences with a high value and decreases as the training proceeds. The neighborhood function maximizes the adaption magnitude of the winning neuron and its closest neighbors and decreases the adaption magnitude of the neurons that are further away in the network. This adaption process repeatedly continues until the training process is completed [58,61]. Due to random procedures included in the SOM analysis (i.e., initialization and entry order of the input vector), this process was iterated 100 times. This increases the odds of finding the best solution to the problem. Finally, 1600 SOMs resulted from the two training algorithms, four neighborhood functions, and two initialization methods (i.e., 100 × 2 × 4 × 2). Topographical and quantization errors (the average Euclidean difference between participants’ input vectors and the weight vectors of the neuron that they are assigned to) were multiplied and the map with the minimum error was selected [61,62]. Subsequently, to categorize neurons into groups according to the values of the input variables (intrinsic motivation, integrated regulation, identified regulation, introjected regulation, external regulation, amotivation, autonomy satisfaction, competence satisfaction, and relatedness satisfaction), a k-means method was used. In order to enable a solution that allows reasonable interpretations by avoiding an excessive amount of clusters, the number of clusters was restricted to range between 2 and 10. The final number of clusters was chosen depending on the Davies–Bouldin index [63] (see Figure 1). The resulting clusters represent typical profiles of adolescents with regard to their AT-related behavioral regulation and BPNs satisfaction. 

Further analyses were conducted with IBM SPSS Statistics 27. Referring to research question 1, differences in the input variables between the identified clusters were tested using Welch’s-ANOVA and the Games–Howell post-hoc test (level of significance *p* < 0.05), due to no homogeneity of variances (Levene’s tests, *p* < 0.05). Effect sizes were calculated using the adjusted omega squared (est. ω^2^) test. To answer research question 2, whether overall AT differed between clusters, differences in the proportion of ways traveled actively were assessed using Welch’s-ANOVA and the Games–Howell post-hoc test (level of significance *p* < 0.05) because the assumption of homogeneity of variance was not met (Levene’s test, *p* < 0.05). Research question 3 addressed whether travel mode and travel distance were associated with cluster membership, when destinations are considered individually. Association of travel mode and cluster membership was assessed using the Chi-Squared test (level of significance *p* < 0.05); Cremér’s V was estimated to indicate effect size according to Cohen [64]. Since adjusted residuals follow a z-distribution, a value of +/−1.96 indicated significant differences between the observed number of adolescents using a travel mode (active or passive) to a destination in a cluster and the expected number. Difference between clusters in distance to school was assessed using Welch’s-ANOVA (level of significance *p* < 0.05) due to the lack of homogeneity of variance (Levene’s test, *p* < 0.05). Association of distance to friends or relatives, shopping facilities, and leisure facilities with cluster membership was assessed using the Chi-Squared test (level of significance *p* < 0.05); Cremér’s V was estimated to indicate effect size according to Cohen [64]. Lastly, to answer research question 4, the Chi-Squared test (level of significance *p* < 0.05) was applied to analyze whether cluster membership was associated with sex/gender. Additionally, differences between the clusters regarding BMI and age were tested by ANOVAs with a significance level of *p* < 0.05.

## 3. Results

### 3.1. Cluster Description

The results of the SOM analysis are presented in Figure 1. For each of the nine input variables, component planes were created (Figure 1A) in which neurons are presented as hexagons. Each adolescent remains in the same neuron in each of the nine component planes. Further, participants located in the same neuron exhibit similar values regarding the input variables. Neurons were grouped into clusters (C1, C2, C3, C4, C5, C6), which are presented in different colors in Figure 1B. The quantization error suggested six clusters (Figure 1C). Figure 1D presents how many adolescents were assigned to each cluster (n_C1_ = 81; n_C2_ = 70; n_C3_ = 69; n_C4_ = 79; n_C5_ = 84; n_C6_ = 134). 

Welch’s-ANOVAs indicated significant difference between clusters in every input variable (intrinsic motivation: F(5, 217.66) = 209.90, *p* < 0.001, est. ω^2^ = 0.67; integrated regulation: F(5, 223.34) = 329.77, *p* < 0.001, est. ω^2^ = 0.76; identified regulation: F(5, 223.83) = 218.63, *p* < 0.001, est. ω^2^ = 0.68; introjected regulation: F(5, 227.40) = 71.59, *p* < 0.001, est. ω^2^ = 0.41; external regulation: F(5, 221.70) = 83.98, *p* < 0.001, est. ω^2^ = 0.45; amotivation: F(5, 208.82) = 72.04, *p* < 0.001, est. ω^2^ = 0.41; autonomy: F(5, 211.84) = 80.34, *p* < 0.001, est. ω^2^ = 0.43; competence: F(5, 198.80) = 29.84, *p* < 0.001, est. ω^2^ = 0.22; relatedness: F(5, 212.04) = 26.69, *p* < 0.001, est. ω^2^ = 20. The means and standard deviations of the nine input variables and for the proportion of ways traveled actively for each of the six clusters are presented in Table 1. Post-hoc comparisons were conducted to examine significant differences in the input variables between the clusters. 

In the following, the profiles of the clusters are described based on the distribution of the six motivational types within each cluster and in consideration of the cluster’s mean scores in the motivational and BPNs satisfaction variables relative to the mean scores of the complete sample (in each variable). 

Cluster 1: Controlled-amotivated—less satisfied. Autonomous forms of motivation and introjected regulation were below the sample’s average scores. All average values of autonomous forms of motivation were lowest in this cluster. External regulation was above the sample’s average and average amotivation was highest in this cluster. The distribution of the motivational types clearly represented a non-self-determined profile. Satisfaction of all BPNs was below the sample’s average.

Cluster 2: Low-autonomously motivated—less satisfied. All autonomous and controlled forms of motivation were below the sample’s average. Amotivation was close to the sample’s average. The distribution of the motivational types was slightly in favor of the autonomous forms of motivation and specifically of intrinsic motivation. Satisfaction of all BPNs was below the sample’s average.

Cluster 3: Mixed moderately motivated—less satisfied. Autonomous forms of motivation were below the sample’s average, while controlled forms and amotivation were above. The distribution of the motivational types was balanced. Satisfaction of competence and relatedness were close, yet still below the sample’s average. Autonomy satisfaction was below the sample’s average.

Cluster 4: Moderate-autonomously motivated—more satisfied. Intrinsic motivation was above the sample’s average, and integrated and identified regulation were close to average. Both controlled forms of motivation and amotivation were below the sample’s average. Average amotivation was second lowest in this cluster. The distribution was in favor of the autonomous forms of motivations. Satisfaction of all BPNs was above the sample’s average.

Cluster 5: Mixed highly motivated—less satisfied. All autonomous forms of motivation were above the sample’s average. Amotivation was close to the sample’s average. The distribution of the motivational types tended towards a self-determined profile; however, controlled forms of motivation were also above the sample’s average. Satisfaction of all BPNs was below the sample’s average.

Cluster 6: High-autonomously motivated—more satisfied. All autonomous forms of motivation were highest in this cluster. Introjected regulation was above the sample’s average, while external regulation was below. Adolescents expressed lowest average amotivation in this cluster. Motivational types were clearly distributed towards a self-determined profile. Satisfaction of all BPNs was above the sample’s average.

An additional bar chart of the mean values of the input variables of the six clusters is provided in Figure 2 to visualize the motivational distribution, amount of motivation, and BPNs satisfaction within each cluster.

### 3.2. Overall Active Travel Behavior

Proportion of ways traveled actively was significantly different between clusters, F(5, 219.78) = 7.85, *p* < 0.001, est. ω^2^ = 0.06. Post-hoc-tests showed that some clusters did not differ significantly from each other in the proportion of ways traveled actively (see Table 1). C1 had the lowest proportion of ways traveled actively compared to all clusters but was only significantly different from C4–C6. C3 did not differ significantly from any other cluster. Adolescents in C5 or C6 reported significantly higher proportions of ways traveled actively compared to adolescents from C1 and C2. 

### 3.3. Travel Mode and Distance to Specific Destinations

Travel mode was significantly associated with cluster membership for travel to school, χ^2^ (5) = 20.23, *p* = 0.001, V = 0.20; travel from school, χ^2^ (5) = 15.63, *p* = 0.008, V = 0.17; to friends or relatives, χ^2^ (5) = 29.43, *p* < 0.001, V = 0.24; travel to shopping facilities, χ^2^ (5) = 13.05, *p* = 0.023, V = 0.16; and travel to leisure facilities, χ^2^ (5) = 23.37, *p* < 0.001, V = 0.22. For travel mode to school, from school, to friends or relatives, and to leisure facilities, adjusted residuals exceeded the threshold of +/−1.96 in C1 and C6. Across those destinations, significantly more adolescents than expected used a passive travel mode in C1, while significantly more adolescents than expected used an active travel mode in C6. Additionally, adjusted residuals exceeded +/−1.96 in C5 for AT to school. Thus, in C5, more adolescents than expected traveled to school actively. Regarding travel mode to shopping facilities, adjusted residuals exceeded the threshold of +/−1.96 only in C2, indicating that significantly more adolescents than expected used a passive travel mode. Across all destinations, a similar trend was observable: the percentage of adolescents traveling actively increases from C1 to C6, while it decreases for passive travel. Detailed information on count, expected count, percentage values, and adjusted residuals for travel mode to the various destinations are presented in the Appendix A.

There was no significant difference in distance to school between clusters, F(5, 213.58) = 1.28, *p* = 0.272. No significant association was found for cluster membership with distance to friends or relatives, χ^2^ (25) = 32.54, *p* = 0.143; with distance to shopping facilities, χ^2^ (25) = 29.68, *p* = 0.236; or with distance to leisure facilities, χ^2^ (25) = 17.01, *p* = 0.881. The distribution of adolescents across distance categories across the six clusters is provided in the Appendix A as well as mean distance to school per cluster (Appendix A).

### 3.4. Cluster Membership and Sex/Gender, Age, and Weight Status

Table 2 shows the respective number of boys and girls as well as mean age and BMI of the adolescents in each cluster. Adolescents’ weight status was assessed in 405 participants. No significant difference appeared between clusters and age, F(5, 511) = 1.49, *p* = 0.192; or BMI, F(5, 399) = 0.70, *p* = 0.627. Further, association of sex/gender and cluster membership, χ^2^ (5) = 7.30, *p* = 0.20 was not significant.

## 4. Discussion

This study aimed to explore naturally occurring profiles in adolescents with regard to motivation and BPNs satisfaction in AT. Using SOM analysis, six clusters were identified based on the six different motivational types (amotivation, external regulation, introjected regulation, identified regulation, integrated regulation, and intrinsic motivation) and the satisfaction of the three BPNs (autonomy, competence, relatedness). The clusters were used to explain differences in AT behavior among adolescents. Additionally, we compared the characteristics (i.e., sex/gender, age, and weight status) of adolescents in each cluster. Before discussing the findings, it is worth reflecting upon the potential problem of reification described by Vansteenkiste and Mouratidis [35] regarding the person-centered approach used. Even though every adolescent is categorized by the psychological characteristics of the specific cluster, it is important to note that some adolescents are more prototypical than others. Thus, cluster membership might better be understood as probability than determination [35]. This should be kept in mind throughout the following discussion of results. 

First, concerning research question 1, the clusters are regarded against the theoretical background of SDT [17,18,20]. The six identified clusters were distinct in the composition of the motivational characteristics. To a varying extend, the motivational pattern for AT in C2, C4, and C6 tended towards an autonomous profile focusing on the self-determined end of the continuum. C6 exhibited the most prototypical profile in terms of an unimodal distribution of motivation for AT [36]. C4 and C2 did not completely align to the proposed simplex structure of motivation [18,37], however, the overall impression was still autonomous (see Figure 2). C1 presented a profile of overall low quantity motivation for AT, which also showed a tendency toward the non-self-determined end of the continuum and thus was classified as controlled. In this case, the particularly high values of amotivation explain the low level of motivation in this profile, because according to definition, amotivation represents the lack of motivation [18]. C3 demonstrated a profile of mixed motivation for AT with comparable levels in all forms of motivation. In C5, the motivational distribution tended towards a more self-determined profile, however, controlled types of motivation were also high. Thus, no clear focus could be identified, leading to the impression of a mixed motivated profile. C3 and C5 exemplify that AT behavior in adolescents could be regulated by autonomous and controlled motivation simultaneously. Overall, our findings suggest the existence of distinct motivational profiles among adolescents regarding AT, including profiles characterized by a mainly autonomous or controlled pattern, as well as profiles with a more heterogeneous motivational distribution. Our findings are supported by research from diverse backgrounds, investigating motivation via person-centered approaches. Throughout different studies, profiles with unimodal curves were identified, as well as profiles that substantiate the heterogeneity of motivation [38,39,40,42,50,65,66]. 

Further, SDT assumes higher satisfaction of the BPNs as precondition for the autonomous forms of motivation [17,18]. In our study, all clusters showed comparatively high levels of BPNs satisfaction towards AT (see Table 1 and Figure 2). Overall, BPNs satisfaction was significantly higher in C4 and C6, both of which demonstrated high values in the autonomous forms of motivation and corresponding low values in the controlled forms and amotivation. This is congruent with the assumptions of SDT that BPNs satisfaction fosters autonomous forms of motivation [17]. Likewise, person-centered studies found that BPNs satisfaction was higher in clusters that expressed higher values in autonomous motivation [41] or were rated as high quality profiles [38,45]. However, C5 also exhibited high values in autonomous motivation but lower satisfaction of the BPNs compared to C4 or C6. The higher values of the controlled forms of motivation in C5 compared to C4 and C6 might yet be explained by the lower BPNs satisfaction in C5. Nevertheless, all clusters demonstrated high levels of BPNs satisfaction and differences between clusters in the BPNs were small (even though sometimes statistically significant, effect sizes were small), especially regarding competence and relatedness. Thus, satisfaction of the BPNs seems to play a minor role for the motivation for AT in adolescents. 

Regarding research questions 2 and 3, the clusters are used to explain the difference in overall AT and travel mode choice to the individual destinations. Concerning BPNs satisfaction in AT, the previously mentioned high values across all clusters limit the explanatory possibilities with regard to AT behavior. Therefore, in the following, the explanation for differences in AT behavior is based primarily on the motivational patterns of the clusters. Nevertheless, it needs to be considered, that variable-based research found a positive association of BPNs satisfaction with AT to and from school [26]. This stresses the need for more person-centered research explicitly focused on BPNs in AT. 

Even though some clusters did not significantly differ in their AT behavior, two beneficial clusters and one disadvantageous cluster were detected. The two most beneficial clusters for overall AT in adolescents constituted C5 (mixed highly motivated—less satisfied) and C6 (high-autonomously motivated—more satisfied), with a significantly higher proportion of active ways than C1 (controlled-amotivated—less satisfied) and C2 (low-autonomously motivated—less satisfied). Both C5 and C6 had similar levels in overall AT. Additionally, when considering the destinations individually, either C6 or C5 contained the highest percentage of adolescents using an active mode across all destinations. For most destinations (except for travel to shopping facilities), C6 exceeded the expected number of adolescents traveling actively. C5 exceeded this number for travel to school. Even though the motivational distribution showed a more autonomous pattern in C6 compared to C5, the higher values of controlled forms of motivation in C5 might not have been harmful because autonomous forms were appropriately present. Indeed, controlled motivation might have complemented autonomous motivation, resulting in a high motivation quantity. This is in line with person-centered research from physical education and leisure-time PA [40,41]. However, in the context of AT, research from Spain and Sweden assumed that adolescents benefit only from autonomous forms of motivation but not from controlled forms [25,29]. This might be attributed to their variable-centered approach, which assumes a homogeneous population of participants and thus, focuses on the effect of motivation rather than on identifying subgroups in a population with similar motivational characteristics. 

Additionally, no significant difference in overall AT behavior was observable between C5 and C6 on the one side and C4 (moderate-autonomously motivated—more satisfied) on the other side. Even though adolescents in C4 exhibited lower levels of autonomous forms of motivation compared to adolescents in C6, C4 still featured an autonomous profile. Nevertheless, overall motivation quantity (i.e., amount of autonomous and controlled motivation) of C4 appeared lower compared to C5 and C6. This might have contributed to the finding that overall AT behavior in C4 was significantly higher than in C1 but not compared to C2. Thus, not only motivation quality (i.e., autonomous pattern) but also quantity (i.e., high autonomous and controlled motivation) was decisive for AT. This is in line with research from physical education, which found that in-class performance was rated equally for students allocated in the autonomously motivated (corresponding to high quality) profile and in the high motivation (corresponding to high quantity) profile [65]. Nevertheless, considering the single destinations, C4 contained a higher percentage of adolescents traveling actively to friends or relatives than C5. This might be attributed to the context of traveling to friends. Presumably, this destination allows more freedom of choice (e.g., when to travel, what to carry) than other destinations. Thus, adolescents might be more likely to decide for their preferred travel mode (e.g., the travel mode for which they are most autonomously motivated). This would indicate the great relevance of a more autonomously motivated and higher satisfied profile in travel contexts with less external constrains. 

The most vulnerable profile is represented by C1 (controlled-amotivated—less satisfied). Adolescents in C1 reported significantly lower overall AT compared to adolescents assigned to C4, C5, or C6. Considering the destinations individually, C1 had the highest percentage of adolescents traveling passively to and from school, to friends or relatives, and to leisure facilities. C1 was classified as controlled-amotivated due to the low values in the more self-determined types of motivation and the corresponding higher values in external regulation and amotivation. The low levels of AT in this cluster are in line with research on students in physical education, which demonstrated lower levels of effort in physical education in more controlled profiles compared to more autonomous profiles [38]. Compared to C5, adolescents allocated to C1 expressed comparable levels of external regulation and even lower values in introjected regulation. Considering the definition of introjected regulation [18], adolescents in this cluster might not yet have internalized external sources of motivation, thus cannot make use of them. Only amotivation was significantly higher in C1 compared to C5. This illustrates the advantage of a profile that expresses a high motivation quantity (i.e., high autonomous and controlled motivation) in combination with low amotivation (i.e., no motivation). 

Interestingly, travel distance to school, to friends or relatives, to shopping facilities, and to leisure facilities was not associated with clusters. Therefore, distance did not seem to be a decisive factor in determining adolescents’ composition of motivation and BPNs satisfaction. This is conflicting with previous research suggesting that among other factors, adolescents’ personal barriers increase with increasing distance [67]. Future research might consider additional factors associated with adolescents’ travel behavior (e.g., general PA, neighborhood walkability, safety concerns) and investigate whether such factors differ between clusters. 

Regarding research question 4, clusters did not differ in terms of sex/gender, age, or BMI. This supports the idea that the profiles found in our study appear equally in male and female adolescents, and across age and BMI groups. However, regarding age, the non-significant difference may result from the small overall age range of the study population. Concerning sex/gender, previous variable-based approaches have found inconsistent results concerning differences in the individual motivational types [29,68]. Further, even though not significant, the proportion of boys was lowest in C1 (identified as most vulnerable cluster for AT) and highest in C6 (identified as one of the most beneficial clusters for AT). Future research might specifically concern the identification of clusters in boys and girls separately to address this issue. Additionally, it needs to be noted that some data were missing regarding BMI, which might have contributed to the non-significant results. Overall, comparison with other studies’ results is difficult due to the diverse settings, variables, and the nature of the person-center approach, which aims to create subgroups from populations. This impedes generalizability of results, thus makes comparisons of studies complicated. Thus, further research, including person-centered analyses, is needed regarding motivation and BPNs satisfaction in AT. 

### 4.1. Implications

Overall, our findings support the applicability of SDT in the context of AT. The cluster solution enabled a better understanding of the SDT-based determinants of AT within adolescents by identifying specifically vulnerable (i.e., adolescents in C1) and resilient (i.e., adolescents in C5 and C6) groups of adolescents. Adolescents who exhibited high quantity of motivation (i.e., high autonomous and controlled motivation) in combination with a tendency towards a higher motivation quality pattern (i.e., a focus towards the self-determined end of the continuum) and highest BPNs satisfaction were most likely to choose an AT mode. Nevertheless, a profile of lower quality due to high levels of controlled motivation seemed to be equally beneficial, as long as autonomous motivations were appropriately present and high values of introjected regulation indicated internalization. Thus, intervention programs should incorporate appropriate behavior-change techniques that aim at creating motivational profiles resembling the profile of C6 (high-autonomously motivated—higher satisfied) or C5 (mixed highly motivated—less satisfied) to enable AT to several destinations in adolescents’ daily life. Even if a profile comparable to C6 might not be achieved, a profile resembling C1 should be avoided in any case. More specifically, the amount of both controlled and autonomous forms of motivation should be increased, which subsequently would involve a reduction of amotivation. Ideally, autonomous motivation should exceed controlled motivation, creating high-quality motivation profiles. However, future research should consider two issues that arise from this finding. First, we found that for destinations offering greater freedom of choice, profiles characterized by a more autonomous distribution of motivation and higher BPNs satisfaction might be more beneficial for choosing an AT mode. Thus, research should put special emphasis on examining diverse destinations that offer varying degrees of freedom to clarify the utility of profiles that differ in their composition of motivation and BPNs satisfaction. This directly relates to the second issue of a potentially poorer quality travel experience, when adolescents face a high number of controlled motivations. Thus, longitudinal studies might contribute to clarify long-term consequences, specifically, whether profiles involving high levels of controlled motivation provide a sustainable way of promoting AT behavior. Taking into account that distance did not differ between clusters highlights the benefits of motivational intervention programs to increase AT to destinations regardless of distance. Further, the association of cluster and travel mode across all examined destinations supports the relevance of the clusters in the context of a variety of destinations in the daily life of youth and not just in the context of school travel. Thus, programs might not necessarily have to be placed in the context of a specific destination, but addressing motivation towards AT in general could enhance AT to several destinations. 

### 4.2. Strength and Limitations

To the best of our knowledge, this was the first study to use a person-centered approach to analyze motivational variables in the context of general AT in adolescence in order to identify typical occurring profiles. A strength of the study was the focus on the adolescent population, which is at risk for physical inactivity and is rather under-researched in the field of AT compared to younger children [54,55]. Assessing AT behavior without focusing exclusively on the school context demonstrated another unique feature of the ARRIVE study compared to most other research in this field [53]. Using a person-centered approach allowed identifying groups of adolescents that share specific combinations of motivational characteristics and helped identify particularly vulnerable adolescents (e.g., C1). Additionally, this analysis acknowledged the assumption that the presence of a single form of motivation does not neglect the absence of any other form [35]. Further, this research benefited from the detailed assessment of intrinsic motivation, integrated regulation, identified regulation, introjected regulation, external regulation, and amotivation, without collapsing them to a single score, which subsequently allowed an understanding of the unique composition within each cluster. Lastly, to assess motivation towards AT and BPNs satisfaction, we made use of questionnaires, which have previously been used and validated in the population of adolescents [25,26,27,28,29]. The limitations of the study included the purposely drawn sample that prevents the generalizability of the results, the exclusive use of self-report data, as well as its cross-sectional design. A longitudinal design could have specifically helped clarify whether the small deviations in BPNs satisfaction contributed to the motivational differences across clusters. Regarding the assessment of motivation and BPNs satisfaction, it needs to be noticed that adolescents did not answer the questionnaires separately for each of the destinations, but once for active travel in general. Future research might consider this and investigate whether these determinants of AT differ when a specific reference to a destination is given. Additionally, AT behavior assessment was based on the usual travel mode which does not take the actual quantity of AT into account. Further, we collapsed several travel modes into a dichotomous variable ‘active’ and ‘passive’ travel behavior. Even though research suggests the analysis of the precise modes (e.g., walking, cycling, car etc.) [69], we aimed to enable a more concise presentation of the already detailed analysis. Regarding the assessment of distance to the analyzed destinations, it needs to be mentioned that distance was not assessed uniformly across destinations. While distance to school was assessed via parents and by a continuous variable, distance to the remaining destinations was assessed via adolescents and by categories. Lastly, it is worth mentioning, that the survey was conducted during the COVID-19 pandemic. Even though data were collected in June 2021 when restrictions were low in Germany, we cannot estimate the extent to which this might have impacted the results. For example, temporary interventions, such as pop-up bicycle lanes, might have temporarily altered travel patterns during the pandemic [70].

## 5. Conclusions

Based on intrinsic motivation, integrated regulation, identified regulation, introjected regulation, external regulation, amotivation, and the satisfaction of autonomy, competence, and relatedness, the SOM analysis revealed six distinct clusters. The clusters helped to explain differences in AT behavior in adolescents. High motivation quality in combination with high quantity of motivation was most beneficial for AT behavior. Additionally, high levels of controlled forms of motivation did not appear harmful with regard to AT behavior as long as the cluster was still in favor of the autonomous forms. However, future research should investigate whether controlled motivation might provide poor quality experiences and thus do not constitute a sustainable way of promoting AT behavior. The most vulnerable cluster was characterized by low motivation quantity and quality in combination with high levels of amotivation. Interpretation of the relevance of BPNs satisfaction was limited, as the values hardly differed between the clusters. Nevertheless, even slightly higher BPNs satisfaction seemed to increase the chance of a more autonomously motivated profile. Associations of clusters with travel mode across all examined destinations highlight the applicability of our findings to a variety of destinations beyond the school destination. Findings regarding distance suggest that interventions to promote AT in the sense of achieving a high-autonomously motivated profile might be successful regardless of distance to a destination.

## Figures and Tables

**Figure 1 behavsci-13-00272-f001:**
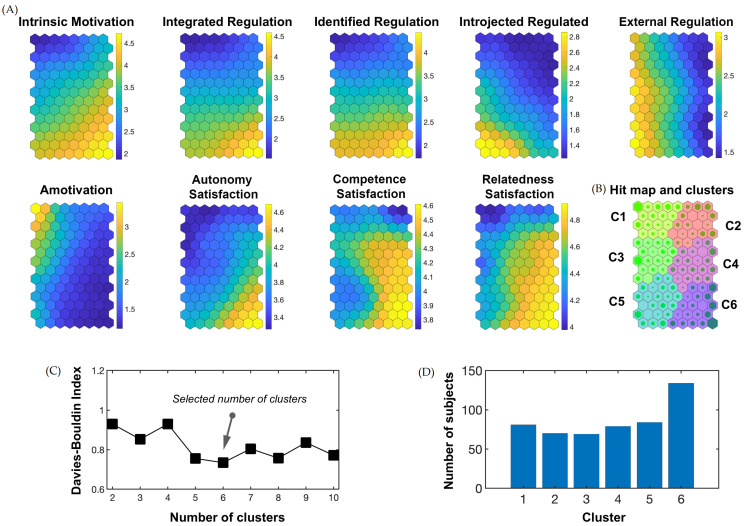
Component planes, clusters (C1–C6), and hits obtained by the SOM analysis. (**A**) Component planes of the nine input variables (intrinsic motivation, integrated regulation, identified regulation, introjected regulation, external regulation, amotivation, autonomy satisfaction, competence satisfaction, and relatedness satisfaction). Yellow neurons indicate comparatively high values; dark blue neurons indicate comparatively low values, depending on the sample’s distribution. (**B**) Hit map with the six clusters (C1–C6). The bigger the green filling of a neuron, the higher the number of adolescents assigned to the neuron. (**C**) Quantization error according to the possible number of clusters selected. (**D**) Overview on number of participants in every cluster.

**Figure 2 behavsci-13-00272-f002:**
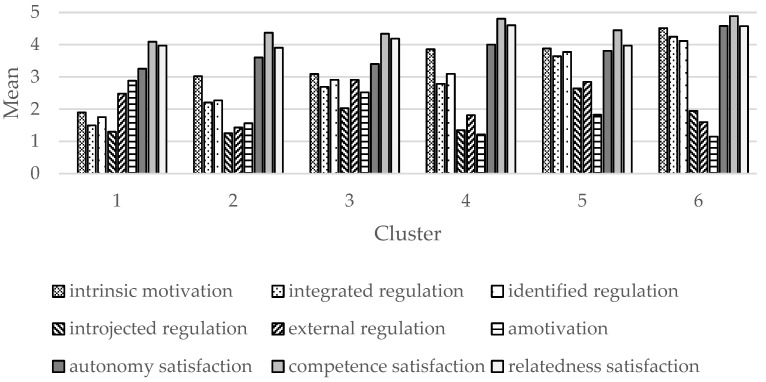
Overview on mean values of SOM input variables across the six clusters.

**Table 1 behavsci-13-00272-t001:** Means (standard deviation) and pairwise comparison of SOM input variables and proportion of ways traveled actively for all six clusters and for the total sample.

	Cluster 1	Cluster 2	Cluster 3	Cluster 4	Cluster 5	Cluster 6	Total
Intrinsic Motivation	1.90 (0.70)	3.02 (0.66) ^3^	3.09 (0.57) ^2^	3.86 (0.58) ^5^	3.88 (0.52) ^4^	4.51 (0.50)	3.51 (1.05)
Integrated Regulation	1.49 (0.46)	2.20 (0.61)	2.69 (0.53) ^4^	2.78 (0.61) ^3^	3.64 (0.50)	4.24 (0.62)	3.01 (1.12)
Identified Regulation	1.75 (0.54)	2.27 (0.64)	2.91 (0.55) ^4^	3.09 (0.56) ^3^	3.77 (0.48)	4.11 (0.66)	3.12 (1.03)
Introjected Regulation	1.30 (0.41) ^2,4^	1.25 (0.30) ^1,4^	2.03 (0.56) ^6^	1.35 (0.39) ^1,2^	2.64 (0.81)	1.94 (0.70) ^3^	1.78 (0.76)
External Regulation	2.48 (0.98) ^5^	1.43 (0.40) ^6^	2.91 (0.72) ^5^	1.81 (0.70) ^6^	2.85 (0.76) ^1,3^	1.60 (0.63) ^2,4^	2.12 (0.92)
Amotiation	2.88 (1.15) ^3^	1.56 (0.62) ^5^	2.52 (0.80) ^1^	1.21 (0.35) ^6^	1.82 (0.98) ^2^	1.15 (0.39) ^4^	1.78 (0.99)
Autonomy Satisfaction	3.25 (0.81) ^2,3^	3.60 (0.72) ^1,3,5^	3.40 (0.66) ^1,2^	4.00 (0.56) ^5^	3.80 (0.48) ^2,4^	4.58 (0.42)	3.87 (0.77)
Competence Satisfaction	4.09 (0.98) ^2,3,5^	4.37 (0.74) ^1,3,5^	4.34 (0.64) ^1,2,5^	4.80 (0.32) ^6^	4.45 (0.57) ^1,2,3^	4.89 (0.23) ^4^	4.54 (0.67)
Relatedness Satisfaction	3.97 (0.85) ^2,3,5^	3.90 (0.72) ^1,3,5^	4.19 (0.69) ^1,2,5^	4.60 (0.45) ^6^	3.97 (0.65) ^1,2,3^	4.57 (0.43) ^4^	4.24 (0.69)
Proportion of ways traveled actively (all destinations)	0.47 (0.34) ^2,3^	0.54 (0.34) ^1,3,4^	0.58 (0.33) ^1,2,4,5,6^	0.65 (0.26) ^2,3,5,6^	0.69 (0.28) ^3,4,6^	0.71 (0.28) ^3,4,5^	0.62 (0.31)

Note: Pairwise comparisons between clusters (Games–Howell post-hoc) were significant (*p* < 0.05) except those marked with superscript numbers (1, 2, 3, 4, 5, 6), which indicate non-significant differences to the respective cluster (e.g., Cluster 1, introjected regulation ^2,4^ = introjected regulation of Cluster 1 is not significantly different from introjected regulation of Cluster 2 or Cluster 4).

**Table 2 behavsci-13-00272-t002:** Description of the clusters by sex/gender, age, and weight status.

	N	Male/Female	Age *	BMI *
Total	517 (100%)	263 (50.9%)/254 (49.1%)	13.11 (1.33)	19.23 (3.32)
Cluster 1	81 (15.7%) ^a^	31 (38.3%)/50 (61.7%) ^b^	13.11 (1.26)	19.55 (3.47)
Cluster 2	70 (13.5%) ^a^	36 (51.4%)/34 (48.6%) ^b^	13.31 (1.46)	19.18 (3.12)
Cluster 3	69 (13.3%) ^a^	37 (53.6%)/32 (46.4%) ^b^	13.06 (1.36)	19.31 (4.04)
Cluster 4	79 (15.3%) ^a^	43 (54.4%)/36 (45.6%) ^b^	12.86 (1.32)	19.27 (2.93)
Cluster 5	84 (16.2%) ^a^	41 (48.8%)/43 (51.2%) ^b^	12.85 (1.33)	18.54 (2.75)
Cluster 6	134 (25.9%) ^a^	75 (56.0%)/59 (44.0%) ^b^	13.17 (1.34)	19.41 (3.45)

* Age and BMI: Data are expressed as mean and standard deviation. ^a^ Percentage refers to total amount of participants (N = 517). ^b^ Percentage of male and female adolescents referring to the total amount of participants assigned to the respective cluster.

## Data Availability

The data presented in this study are available on request from the corresponding author.

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
