# Peer review of "Motivation and Basic Psychological Needs Satisfaction in Active Travel to Different Destinations: A Cluster Analysis with Adolescents Living in Germany"

_behavsci, 2023, doi:10.3390/bs13030272_

Round 1

Reviewer 1 Report

The topic taken up is very interesting. There are not many studies devoted to this topic. The material is well prepared. A rich theoretical introduction explains the undertaking of this research problem.

I think that the material presented for review could be improved by taking into account the following suggestions.

When describing the research tool, it would be worth specifying what answers, according to the Likert's scale, the respondents had to choose from.

Describing the statistical methods, I would suggest specifying which factors were included in the cluster analysis.

Reviewer 2 Report

This is a very interesting manuscript that examines the motivation of adolescents and their psychological needs satisfaction in active travel to different destinations, with the use of cluster analysis.

Given that mobility habits are too difficult to brake (https://doi.org/10.1186/s12544-018-0340-6) the study of the motivation and the factors that affect these habits is crucial to understand how we could modify people’s mobility behavior, especially the adolescents’.

The manuscript is well-written and includes all the required information. Nevertheless, I would expect to see in the title the fact that the survey was conducted in Germany since mobility behavior might be different from country to country.

In the discussion, I would like to see if the research found any differentiations in active mobility patterns due to the COVID-19 pandemic, given social distancing measures and the temporary interventions that promoted active mobility such as e.g., the pop-up bicycle lanes (e.g., see  https://doi.org/10.1088/1755-1315/899/1/012057).

The finding that there was no significant difference in distance to school between clusters is indeed surprising and unexpected. Are there any insights on whether this is due to the fact that the participants were adolescents that are much more active compared to the general population or if there are active mobility schemes for commuting to and from school in a place that facilitate the trips and, therefore, the distance becomes less significant? I have in mind schemes like the “Walking Bus” and/or “Cycling Bus” (e.g., see https://doi.org/10.1016/B978-0-08-102671-7.10183-6).

Perhaps the questions above were not included in the research questions, but still, I think these are interesting and important and could be proposed as guidelines for further research. 
